# Age-related decline in blood-brain barrier function is more pronounced in males than females in parietal and temporal regions

Xingfeng Shao[1]*, Qinyang Shou[1], Kimberly Felix[2], Brandon Ojogho[1], Xuejuan Jiang[2,3], Brian T Gold[4], Megan M Herting[2], Eric L Goldwaser[5,6], Peter Kochunov[7], Elliot Hong[7], Ioannis Pappas[1], Meredith Braskie[1], Hosung Kim[1], Steven Cen[8], Kay Jann[1], Danny JJ Wang[1,8]*

[1]USC Stevens Neuroimaging and Informatics Institute, Keck School of Medicine, University of Southern California, Los Angeles, United States; [2]Department of Population and Public Health Sciences, Keck School of Medicine, University of Southern California, Los Angeles, United States; [3]Department of Ophthalmology, Keck School of Medicine, University of Southern California, Los Angeles, United States; [4]Department of Neuroscience, College of Medicine, University of Kentucky, Frankfort, United States; [5]Maryland Psychiatric Research Center, Department of Psychiatry, University of Maryland School of Medicine, Baltimore, United States; [6]Interventional Psychiatry Program, Department of Psychiatry, Weill Cornell Medicine, New York, United States; [7]Louis A. Faillace Department of Psychiatry and Behavioral Sciences at McGovern Medical School, The University of Texas Health Science Center at Houston, Houston, United States; [8]Department of Radiology and Neurology, Keck School of Medicine, University of Southern California, Los Angeles, United States

**\*For correspondence:**
xshao@ini.usc.edu (XS);
jj.wang@loni.usc.edu (DJJW)

**Competing interest:** The authors declare that no competing interests exist.

## eLife assessment

This study presents a **valuable** finding that the blood-brain barrier functionality changes with age and differs between males and females. The analysis is **solid**, comprising a large and racially diverse dataset, and utilizes a contrast-agent-free MRI method. Since limited work has been done in the MRI field on the blood-brain barrier using this method, this study is of great interest to neuroimaging researchers and clinicians.

**Abstract** The blood-brain barrier (BBB) plays a pivotal role in protecting the central nervous system (CNS), and shielding it from potential harmful entities. A natural decline of BBB function with aging has been reported in both animal and human studies, which may contribute to cognitive decline and neurodegenerative disorders. Limited data also suggest that being female may be associated with protective effects on BBB function. Here, we investigated age and sex-dependent trajectories of perfusion and BBB water exchange rate (kw) across the lifespan in 186 cognitively normal participants spanning the ages of 8–92 years old, using a non-invasive diffusion-prepared pseudo-continuous arterial spin labeling (DP-pCASL) MRI technique. We found that the pattern of BBB kw decline with aging varies across brain regions. Moreover, results from our DP-pCASL technique revealed a remarkable decline in BBB kw beginning in the early 60 s, which was more pronounced in males. In addition, we observed sex differences in parietal and temporal regions. Our

findings provide in vivo results demonstrating sex differences in the decline of BBB function with aging, which may serve as a foundation for future investigations into perfusion and BBB function in neurodegenerative and other brain disorders.

## Introduction

The BBB, an intricate network of endothelial cells, pericytes, basement membrane, and astrocyte end-feet, forms a selective permeability shield between the CNS and blood. This dynamic interface regulates the passage of macromolecules to allow the passage of nutrients and prevent the entry of potential neurotoxins and pathogens (*Sweeney et al., 2019*). There is a natural decline and break-down of BBB function with aging as revealed by both animal and human studies (*Erdő et al., 2017*; *Knox et al., 2022*). When combined with a second hit such as inflammation and/or ischemia, this trend of declining BBB function with healthy aging may become more detrimental leading to neuronal damage and pathogenesis of a variety of neurological disorders including multiple sclerosis (*Wengler et al., 2020*; *Ingrisch et al., 2012*), stroke (*Tiwari et al., 2017*; *Villringer et al., 2021*; *Wardlaw et al., 2003*), traumatic brain injury (*Cash and Theus, 2020*), Parkinson's Disease (*Ivanidze et al., 2020*), cerebral small vessel disease (cSVD) (*Wardlaw et al., 2003*; *Bridges et al., 2014*) and Alzheimer's disease (*Lin et al., 2021*; *Zlokovic, 2011*; *Montagne et al., 2017*). Considering the prominent difference in the prevalence of these disorders between males and females, it is reasonable to speculate that there may be sex differences in BBB integrity which may vary with age. A few animal studies have reported a protective effect of female sex hormones on BBB permeability, with ovariectomized rats showing increased Evan's blue dye extravasation into the brain which was normalized by estrogen replacement (*Cipolla et al., 2009*; *Wilson et al., 2008*). Two human studies showed that compared to males, females have significantly lower CSF/serum albumin ratio and reduced BBB permeability to Gadolinium (Gd)-based contrast agent using dynamic contrast enhanced (DCE) MRI (*Parrado-Fernández et al., 2018*; *Moon et al., 2021*). However, lumber puncture for CSF sampling and venous injection of Gd-based contrast agent are required to measure BBB permeability in existing human studies. However, these methods have drawbacks as the CSF/serum albumin ratio does not provide regional information and DCE MRI is not suitable for specific populations (e.g. children and participants with renal dysfunction).

Recently, a non-invasive MRI technique termed diffusion-prepared pseudo-continuous arterial spin labeling (DP-pCASL) (*St Lawrence et al., 2012*; *Shao et al., 2019*) has been proposed for mapping BBB water exchange rate (kw) by differentiating labeled water in the cerebral capillaries and brain tissue with appropriate diffusion weighting. Compared to Gd-based contrast agents, water is an endogenous tracer with a small molecular weight (18 Da), and water exchange across the BBB occurs at a relatively high level and is mediated by passive diffusion, active co-transport through the endothelial membrane, and facilitated diffusion through the dedicated water channel, aquaporin-4 (AQP4), at the end-feet of astrocytes (*Papadopoulos and Verkman, 2013*; *Ohene et al., 2019*). As a result, assessing kw, which may reflect the function of AQP4 as suggested by preclinical studies (*Ohene et al., 2019*), is expected to provide a more sensitive metric of BBB function compared to DCE MRI.

DP-pCASL has recently been applied in a range of brain disorders and has been validated by mannitol administration to open the BBB and use an ischemia-reperfusion model to disrupt BBB in rats (*Tiwari, 2015*; *Tiwari et al., 2017*). As opposed to increased BBB leakage detected by DCE MRI (*Shao et al., 2020*), reduced BBB kw has been found in a wide range of CNS disorders including obstructive sleep apnea (*Palomares et al., 2015*), schizophrenia (*Goldwaser et al., 2023*), multiple sclerosis (*Wengler et al., 2020*), and heredity cSVD (*Li et al., 2023*; *Ling et al., 2023*). In particular, emerging evidence suggests a high relevance between BBB water exchange and glymphatic function. Amyloid plaques may interfere with the normal function of AQP-4 and lead to its depolarization (*Rosu et al., 2020*; *Yang et al., 2011*). Recent studies reported that BBB kw was lower in those with more *APOE* ε4 alleles and more amyloid deposition in the brain as detected by PET (*Uchida et al., 2022*). Similarly, in cognitively normal adults, BBB kw was lower in those with lower CSF Aβ–42 (typically associated with the more pathological condition) (*Gold et al., 2020*). These data suggest associations between reduced BBB water exchange, compromised glymphatic function, and impaired clearance of Aβ (*Silva et al., 2021*), potentially due to dysfunctional AQP4 (*Yang et al., 2011*; *Escartin et al., 2021*). Reduced BBB kw with aging has also been reported in small groups of young and aged persons

(*Anderson et al., 2020*; *Ford et al., 2022*; *Zachariou et al., 2024*), suggesting that the glymphatic system becomes less effective in older adults (*Benveniste et al., 2019*).

Despite the promising findings, the dynamic spatiotemporal pattern of BBB kw evolution with age and sex is not yet understood. The aim of the present study was to investigate the potentially region-specific trajectory of kw variations with age and sex in a cohort of 186 cognitively normal participants aged from 8 to 92 years. We recruited only cognitively normal individuals to discern the influence of natural aging on BBB function, which may serve as a future foundation for detecting BBB aberrations in neurodegenerative and other brain disorders. Based on the literature and our recent findings, we hypothesized that age-related BBB kw declines may be more pronounced in males than females.

## Results

### Population

This study included 186 participants aged 8–92 years (89 males and 97 females) with a diverse racial distribution including 65 White/Caucasian (29 males, 36 females), 27 Latinx (11 males, 16 females), 47 African American (21 males, 26 females), and 47 Asian (28 males, 19 females) (*Figure 1—figure*

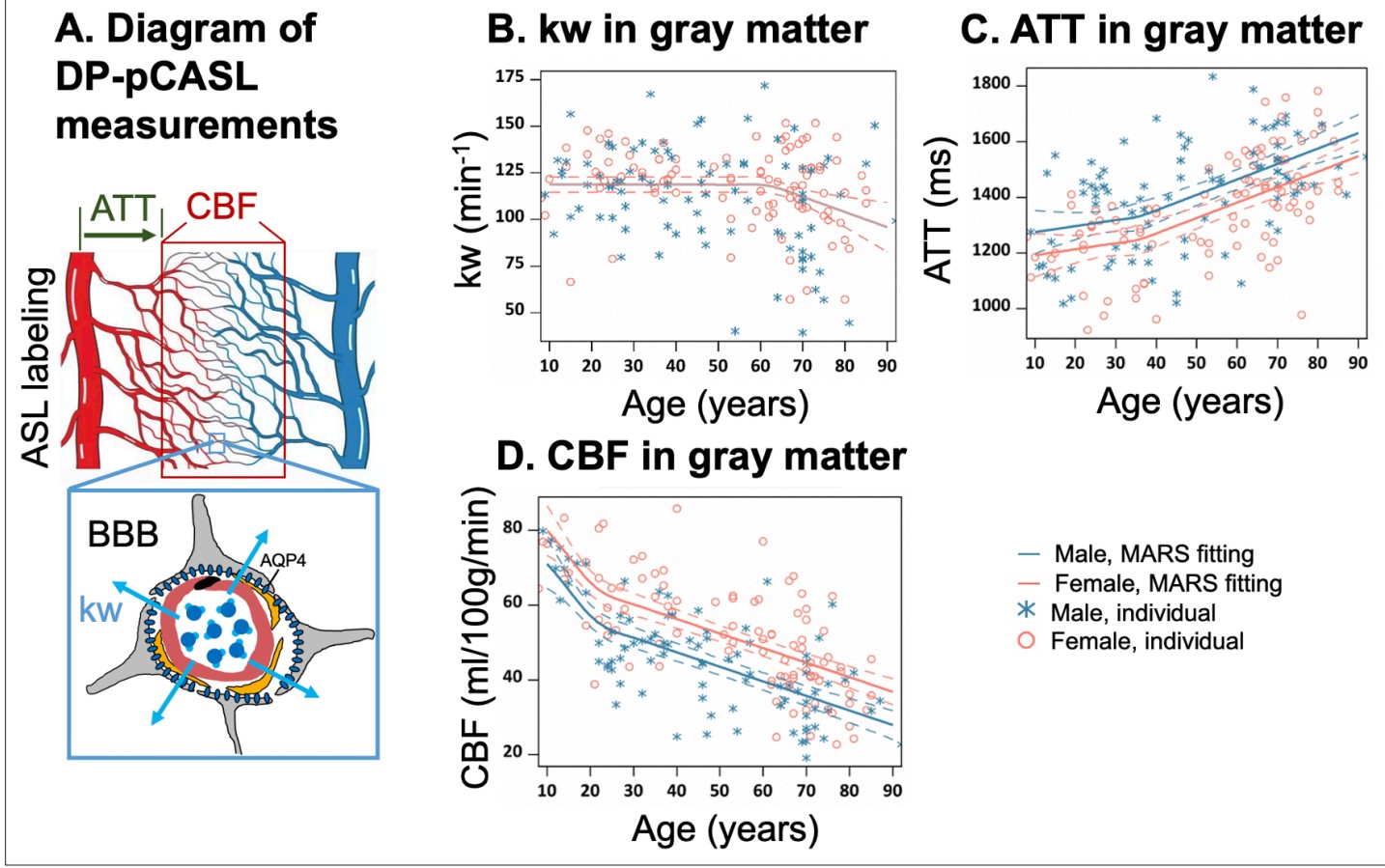

**Figure 1.** Illustration of diffusion prepared pseudo-continuous arterial spin labeling (DP-pCASL) measurements and age-related trends in kw, arterial transit time (ATT), and cerebral blood flow (CBF) in gray matter. (**A**) Diagram illustrating the DP-pCASL measurements. ATT represents the transit time of the labeled blood traveling from DP-pCASL labeling plane to the imaging voxel, CBF refers to the amount of labeled blood supplied to the brain per unit of time (or perfusion), and kw describes the rate of blood water exchanging from intravascular space (capillaries) into extravascular space (tissue) facilitated by multiple transport mechanisms including AQP-4 water channel assisted transport. (**B–D**) Scatter plots representing age-related distribution of kw (**B**), ATT (**C**) and CBF (**D**) values in the gray matter. In all scatter plots, individual data points for males and females are indicated by blue asterisk symbols and red circles, respectively, while the corresponding MARS fitting curves and 95% confidence interval for expected value at each age point are presented as continuous lines and dashed lines in matching colors.

The online version of this article includes the following figure supplement(s) for figure 1:

**Figure supplement 1.** Participant age distribution by sex.

*supplement 1*). Since race may influence the changes in perfusion and kw with aging, it was included as a covariate in the following statistical analyses.

## Age and sex-dependent trajectories of regional kw, ATT, and CBF changes

The DP-pCASL technique provided concurrent measurements of arterial transit time (ATT), cerebral blood flow (CBF), and kw (*Figure 1A*) with a single scan. Anticipating that associations of regional CBF, ATT, kw with age may be nonlinear, especially if BBB degradation initiates within a specific age range, we employed Multivariate Adaptive Regression Splines (MARS) (*Friedman, 1991*) analysis, which is adept at automatically identifying nonlinearities (splines) and interactions among variables with a 10-fold generalized cross validation (GCV) to prevent over-fitting.

Within the gray matter, we observed that BBB kw remained relatively consistent before the age of 62 years and declined thereafter with a yearly slope of –0.82 (95% CI: [–1.35, –0.28]) (p=0.003), while no statistically significant sex differences were observed by MARS analysis (*Figure 1B*, *Supplementary file 1a*). In contrast, ATT was overall longer in males, while CBF was higher in females across the lifespan (*Figure 1C and D*, *Supplementary file 1b and c*). It is worth noting that the patterns

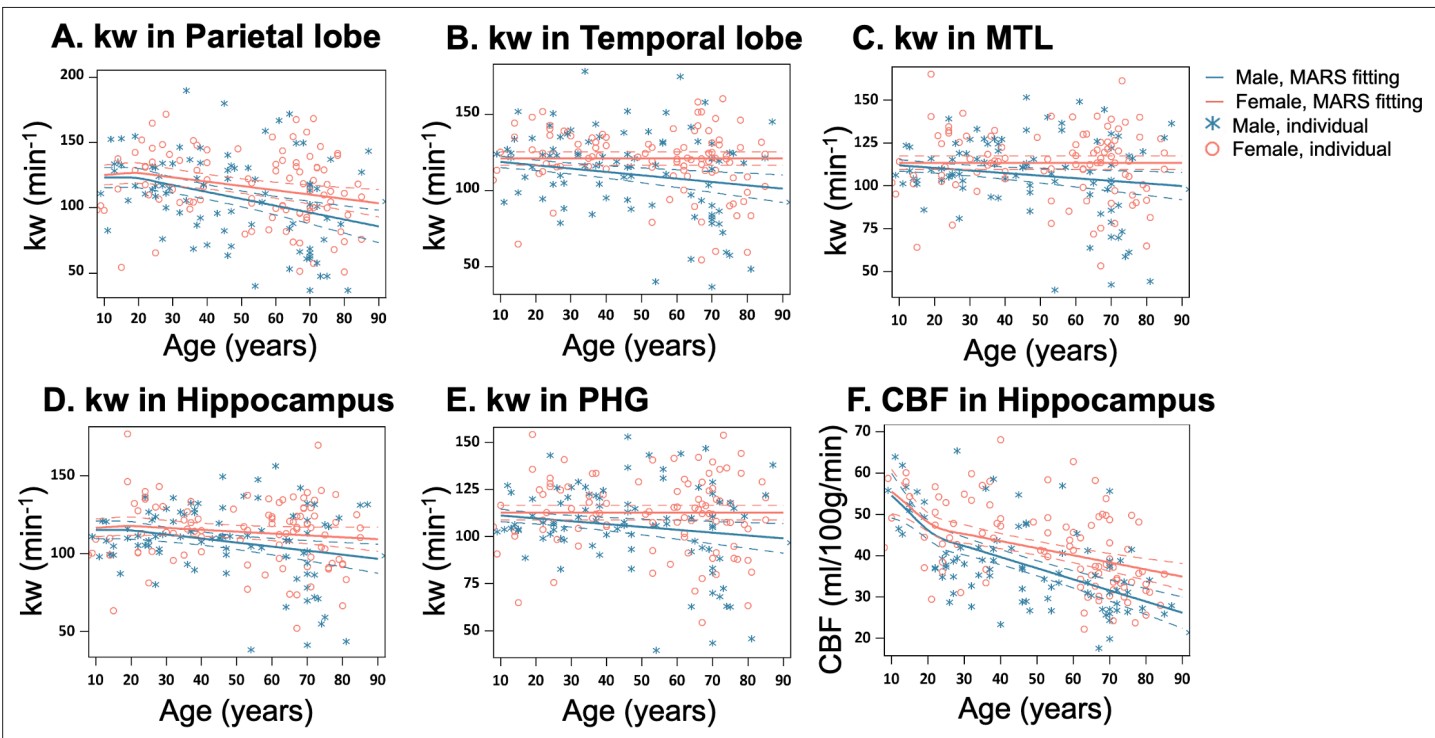

**Figure 2.** Sex-specific age trends in kw and cerebral blood flow (CBF). (**A–E**) Scatter plots representing the age-related distribution of kw values in the parietal lobe (**A**), temporal lobe (**B**), MTL (**C**), hippocampus (**D**) and PHG (**E**). (**E**) Scatter plots representing an age-related distribution of CBF values in the hippocampus. In all scatter plots, individual data points for males and females are indicated by blue asterisk symbols and red circles, respectively, while the corresponding MARS fitting curves and 95% confidence interval for expected value at each age point are presented as continuous lines and dashed lines in matching colors. Abbreviation: MTL, medial temporal lobe; PHG, parahippocampal gyrus.

The online version of this article includes the following figure supplement(s) for figure 2:

**Figure supplement 1.** Regional MARS analysis to reveal the age and sex-dependent trajectories of kw in gray matter (GM), WM, Frontal lobe, Temporal lobe, Parietal lobe, Anterior Cingulate Cortex (ACC), Posterior Cingulate Cortex (PCC), Precuneus, Putamen, Caudate, Amygdala, Hippocampus, Parahippocampal gyrus (PHG), and Mediotemporal lobe (MTL).

**Figure supplement 2.** Regional MARS analysis to reveal the age and sex-dependent trajectories of cerebral blood flow (CBF) in gray matter (GM), WM, Frontal lobe, Temporal lobe, Parietal lobe, Anterior Cingulate Cortex (ACC), Posterior Cingulate Cortex (PCC), Precuneus, Putamen, Caudate, Amygdala, Hippocampus, Parahippocampal gyrus (PHG), and Mediotemporal lobe (MTL).

**Figure supplement 3.** Regional MARS analysis to reveal the age and sex dependent trajectories of arterial transit time (ATT) in gray matter (GM), WM, Frontal lobe, Temporal lobe, Parietal lobe, Anterior Cingulate Cortex (ACC), Posterior Cingulate Cortex (PCC), Precuneus, Putamen, Caudate, Amygdala, Hippocampus, Parahippocampal gyrus (PHG), and Mediotemporal lobe (MTL).

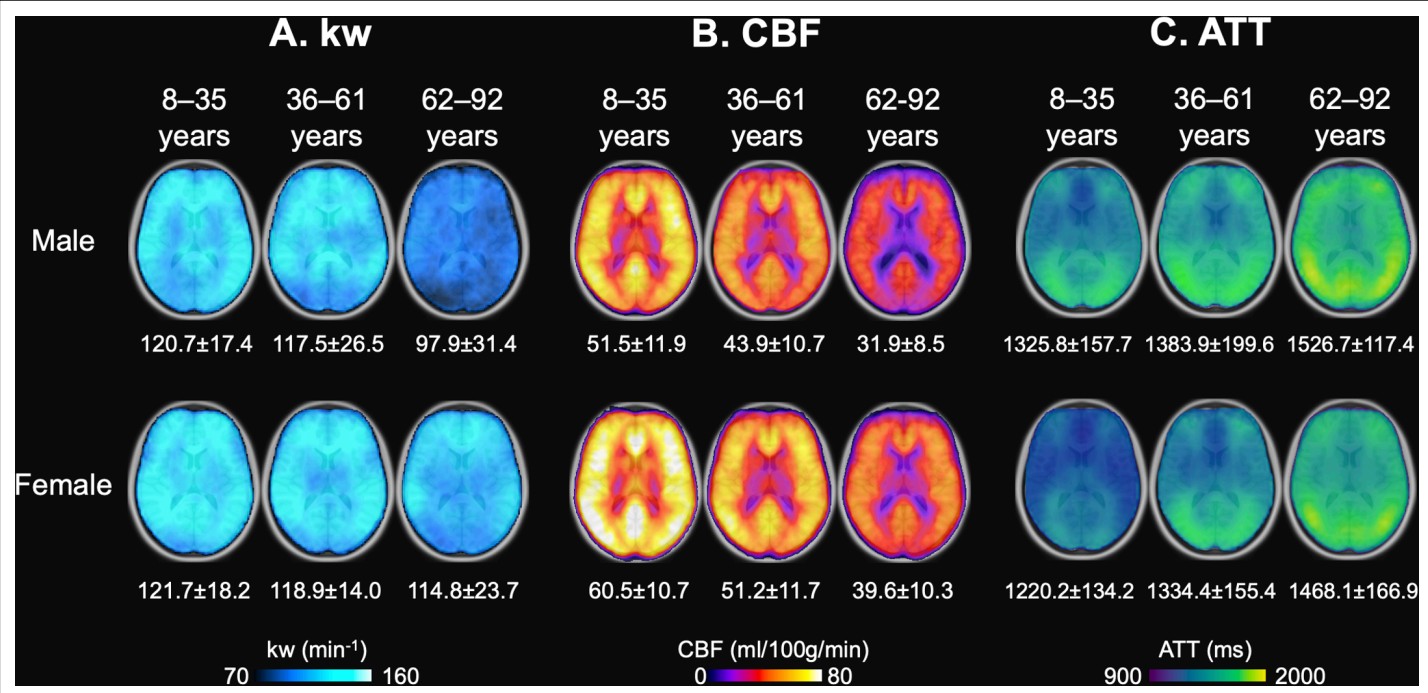

**Figure 3.** Age and Sex-based Variations in kw, arterial transit time (ATT), and cerebral blood flow (CBF). (**A**) kw maps, (**B**) CBF maps, and (**C**) ATT maps, average across three age groups: 8–35 years (Males: n=31, average age 23.0 years; Females: n=26, average age 22.7 years), 36–63 years (Males: n=28, average age 46.8 years; Females: n=28, average age 51.3 years), and 62–92 years (Males: n=30, average age 72.7 years; Females: n=43, average age 72.8 years). These maps are superimposed on T1w anatomical images. Corresponding average kw, CBF, and ATT values are provided beneath each map. Across the spectrum, kw values remained relatively consistent between males and females, though a marked reduction in kw can be observed in males aged 62–92 years. Patterns in the maps suggest an age-related a crease in CBF and increase in ATT, while males generally had lower CBF and longer ATT values compared to females.

The online version of this article includes the following figure supplement(s) for figure 3:

**Figure supplement 1.** Voxel-wise distributions of arterial transit time (ATT) and kw for participants aged over 62 years.

of change in these three physiological parameters were different, with kw showing a turning point around the age of 62 years (*Figure 1B*, *Supplementary file 1a*), ATT around the age of 36 years (*Figure 1C*, *Supplementary file 1c*), and CBF around the age of 22 years (*Figure 1D*, *Supplementary file 1b*). Regional MARS analysis results for all 14 subregions can be found in *Figure 2—figure supplements 1–3* and *Supplementary file 1a, b, c* for kw, ATT, and CBF, respectively.

Sex difference became evident for kw in MARS analyses of brain regions such as the parietal lobe (*Figure 2A*, p=0.01), temporal lobe (*Figure 2B*, p=0.001), medial temporal lobe (MTL) (*Figure 2C*, p=0.008), hippocampus (*Figure 2D*, p=0.02) and parahippocampal gyrus (PHG) (*Figure 2E*, p=0.006) (*Supplementary file 1a*). In these regions, kw decline with age was more pronounced in males than females. In contrast, the rates of CBF decrease and ATT increase with aging were largely consistent between the males and females, except for a more rapid CBF decrease in males within the hippocampus (*Figure 2F*, p<0.001, *Supplementary file 1b*).

## Average kw, ATT, and CBF maps across age groups

We further divided the cohort of 186 participants into three age groups with the similar number of participants spanning a comparable age range, and calculated average maps of kw, CBF, and ATT for the three age groups: children too young adulthood (8–35 years, N=56), middle age (36–61 years, N=55), and older age (62–92 years, N=75) (*Grady et al., 2006*; *Figure 3*). The threshold of 62 years as the starting point for the older age group was consistent with the MARS results, which indicated a notable decline in BBB kw beginning after the age of 62 years. Whole-brain average kw values for males were 120.7±17.4, 117.5±26.5, and 97.9±31.4 min$^{-1}$ across the three age groups, while females' kw values were 121.7±18.2, 118.9±14.0, and 114.8±23.7 min$^{-1}$, respectively. Whole-brain

average CBF values were 51.5±11.9, 43.9±10.7, and 31.9±8.5 ml/100 g/min for males, and 60.5±10.7, 51.2±11.7, and 39.6±10.3 ml/100 g/min for females from young to middle age to elderly groups. Whole-brain average ATT values were 1325.8±157.7, 1383.9±199.6, and 1526.7±117.4 ms for males, and 1220.2±134.2, 1334.4±155.4, and 1468.1±166.9 ms for females across the three age groups. These average maps of three age groups are highly consistent with the regional MARS analysis.

As age progresses, we observed an increase in ATT and a concurrent decrease in CBF, while males had 8.6%, 3.7% and 3.9% longer ATT and 14.8%, 14.3%, and 19.4% lower CBF as compared to females across three age groups respectively (p<0.001, one-way ANOVA). The sex difference in kw values was more pronounced among older participants (62–92 years), while males had an average of 14.7% lower kw compared to females (p<0.001, one-way ANOVA) (*Park, 2009*). Minimal differences (<2%) in mean kw values were found in young and middle-aged cohorts. *Figure 3—figure supplement 1A* shows that in the older participants, males had slightly longer ATT (3.9%, p<0.001) compared to females. However, sex differences can be observed in kw distributions (*Figure 3—figure supplement 1B*), where females show higher mean kw values while males present a broader distribution. To determine whether the lower kw observed in males may be attributable to prolonged ATT, we conducted simulations recalculating females' kw values to match males' ATT by adding 60ms, leading to marginally higher kw values in females (*Figure 3—figure supplement 1C*). These findings suggest that the kw differences observed between males and females are not predominantly driven by ATT differences. Furthermore, considering our SPA model's kw quantification algorithm, which utilizes the ratio of perfusion signals with and without diffusion weightings (*Shao et al., 2019*), it is unlikely that CBF variations contribute significantly to the sex differences in kw. To account for potential confounding effects, we included both ATT and CBF as covariates in the following regression analyses of kw with age and sex.

## Voxel-level analysis of age and sex effects on kw, CBF, and ATT

To investigate the detailed spatiotemporal pattern of perfusion and kw variations with age and sex, we further applied voxel-wise analysis using two generalized linear regression models (GLM) to study the effect of age and the interaction between age and sex on CBF, ATT, and kw respectively (see details in Materials and Methods). In particular, for voxel-wise analysis of each of the three parameters (e.g.

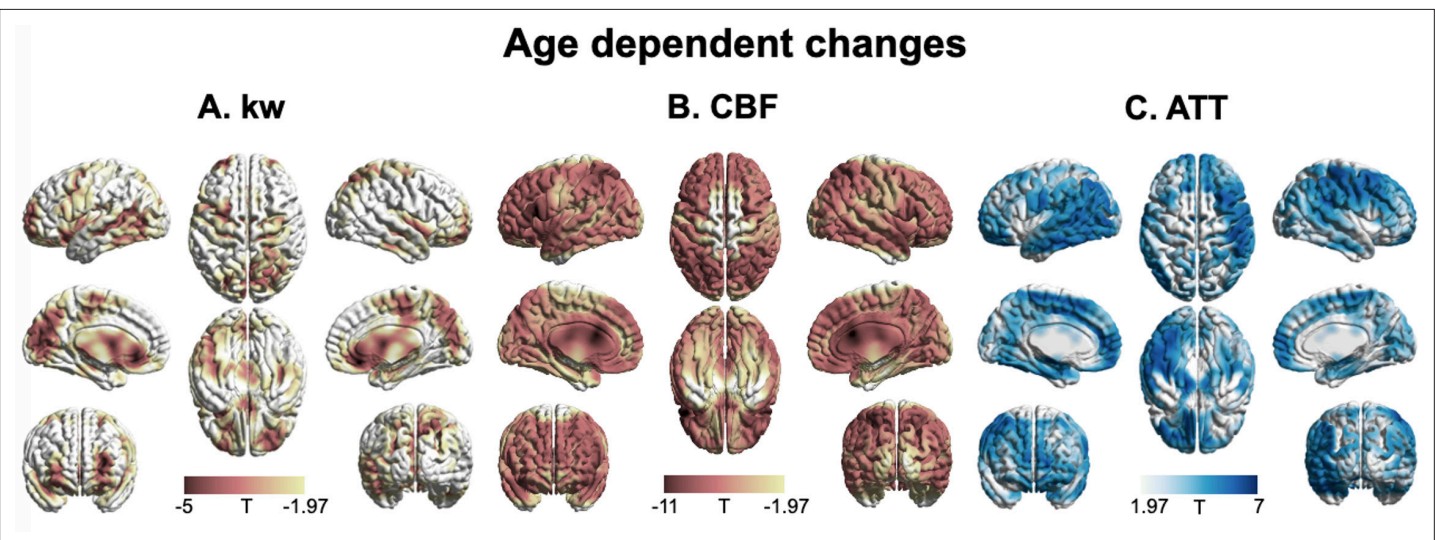

**Figure 4.** 3D renderings of T maps depicting age-related differences. Highlighted areas indicate significant age-related decreases in kw (**A**), decreased cerebral blood flow (CBF) (**B**), and increased arterial transit time (ATT) (**C**). The most pronounced decrease in kw was observed in the lateral and medial prefrontal cortices, anterior cingulate cortex (ACC), posterior cingulate cortex (PCC), temporal lobe, parietal lobe, occipital lobe, and insula. A broad range of brain regions exhibits both CBF reduction and ATT increase. The color scale represents T values, and clusters consisting of over 501 voxels with an absolute T value greater than 1.97 are considered significant and displayed in the figure. The limited slice coverage of diffusion prepared pseudo-continuous arterial spin labeling (DP-pCASL) (96 mm) may account for the absence of detected effects in the upper and lower regions of the brain. The effects of decrease or increase were represented by warm colors (yellow to red) and cold (gray to blue) colors, respectively.

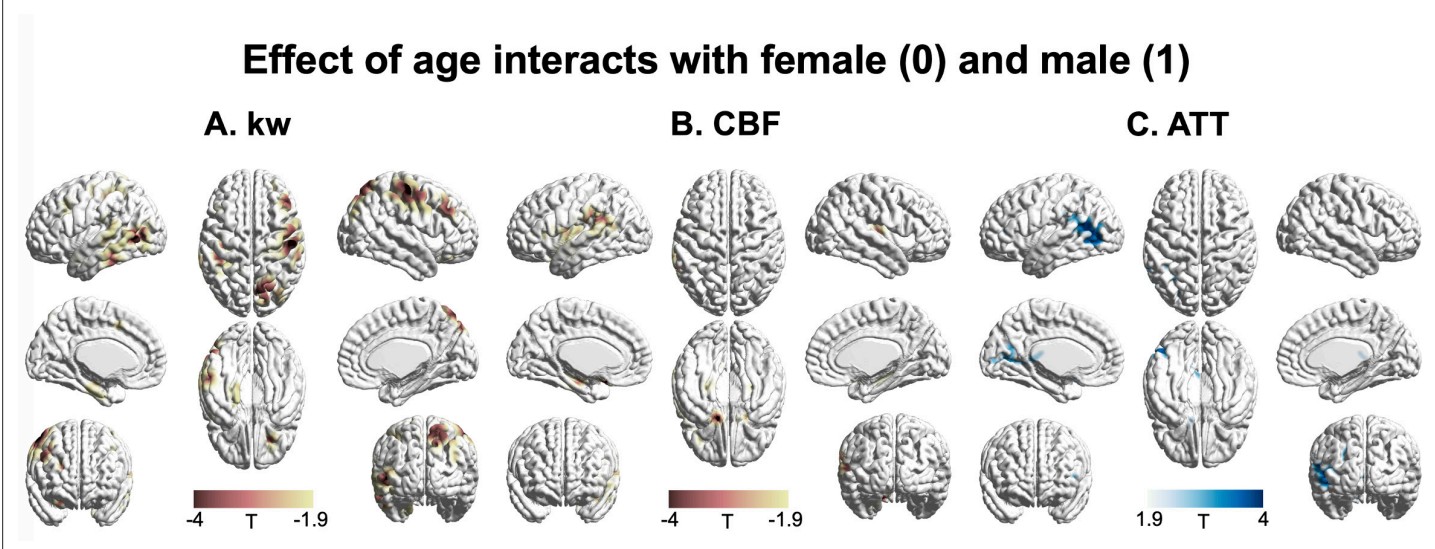

**Figure 5.** 3D renderings of T maps illustrating the interaction effects of age with sex. Highlighted areas indicate: (**A**) Accelerated decrease in kw with aging in males compared to females, most evident in the lateral prefrontal cortex, parietal, and lateral and medial temporal areas. (**B**) Accelerated decrease in cerebral blood flow (CBF) with aging in males compared to females, prominently observed in the supramarginal gyrus, hippocampus, and frontal areas. (**C**) Accelerated increase in arterial transit time (ATT) with aging in males compared to females, with marked changes in supramarginal gyrus, posterior temporal lobe, and calcarine sulcus. The distinct interaction patterns between age and sex across kw, CBF, and ATT can be observed. The color scale denotes T values. Clusters comprising over 501 voxels and possessing an absolute T value exceeding 1.90 are deemed significant and showcased in the figure. The effects of decrease or increase were represented by warm colors (yellow to red) and cold (gray to blue) colors, respectively.

kw), the rest two parameters (e.g. CBF and ATT) were included as covariates in the GLM to control for the potential confounding effects of the three parameters concurrently measured by DP-pCASL.

### Age trend

*Figure 4* shows T-statistic maps of effects (GLM1), which suggested that age-related differences in CBF and ATT were widespread, encompassing the majority of brain regions (*Figure 4B and C*). Conversely, significant decreases in kw were primarily observed in specific brain regions (*Figure 4A*) including the lateral and ventromedial prefrontal cortex, cingulate cortex (CC), precuneus, medial and lateral temporal lobe, occipital lobe, and insula (*Supplementary file 1d*). In the brain regions showing significant age-related kw decreases (*Figure 4A*), these decreases are mostly accompanied by CBF decreases (*Figure 4B*) and ATT increases (*Figure 4C*).

### Sex effect

*Figure 5* displays T-statistic maps showing the interaction effects between age and sex on kw (**A**), CBF (**B**), and ATT (**C**), respectively. We found an accelerated decline in kw as males age in comparison to females in regions including the lateral prefrontal cortex, parietal as well as lateral and medial temporal lobes (*Supplementary file 1e*). Although significant age-dependent changes of CBF and ATT were found in the majority of brain regions, very few clusters remained significant when examining the interaction effects between age and sex. We found a pronounced decline in CBF with advancing age in males relative to females in specific clusters located around the supramarginal gyrus, hippocampus, and frontal lobe (*Supplementary file 1e*). An accelerated increase in ATT with aging in males was found in a few clusters within regions including supramarginal gyrus, posterior temporal lobe, and calcarine sulcus (*Supplementary file 1e*).

### Discussion

To the best of our knowledge, the present study is the first to simultaneously investigate the spatio-temporal trajectories of CBF, ATT, and BBB kw variations with age and sex in a diverse group of cognitively normal participants across the lifespan (*Goldwaser et al., 2023*; *Gold et al., 2020*), enabled by

the innovative DP-pCASL technique. Our findings indicate a pervasive increase in ATT and a decrease in CBF across brain regions (*Figure 4*), with similar rates of change between males and females in the majority of brain regions (*Figures 2 and 5*). Females show higher CBF and shorter ATT compared to males of similar age, which is consistent with literature findings on CBF and ATT variations with age and sex (*Parkes et al., 2004*; *Liu et al., 2012*).

Longer ATT suggests a delayed delivery of blood to brain tissue, possibly due to multiple factors such as compromised vascular elasticity or cerebrovascular reactivity (*Tarumi and Zhang, 2018*). Females have demonstrated relatively higher CBF values compared to males, potentially due to the protective effects of estrogen on the vasculature as well as lower hematocrit levels in females (*Gur et al., 1991*). Furthermore, we found a more pronounced age-related decline in CBF in the hippocampus of males compared to females (*Figure 2*, *Figure 2—figure supplement 1b*). To the best of our knowledge, no study has previously reported this accelerated hippocampal CBF decline in males. This finding may be linked to the accelerated hippocampal volume loss in males, as reported in a study analyzing 19,793 generally healthy UK Biobank participants (*Nobis et al., 2019*). Lower hippocampal perfusion has been associated with poor memory performance (*Rane et al., 2013*; *Heo et al., 2010*), suggesting that males might be more vulnerable to potential cognitive decline (*Gannon et al., 2019*).

We observed a distinct trajectory of kw changes with aging compared to CBF and ATT. To study the potential regional associations between kw and CBF and ATT, we conducted linear regressions between regional kw and regional CBF or ATT, incorporating sex as a covariate, for participants aged 8–61 years and 62–92 years (when BBB kw starts declining), respectively. The results are shown in *Supplementary file 1f*. BBB kw was significantly negatively associated with CBF in the putamen, amygdala, hippocampus, PHG, and MTL in participants aged 8–61 years (when kw was relatively consistent across ages), but no significant correlations were found in any brain regions in the 62–92 years group. In contrast to CBF, kw was significantly negatively associated with ATT in the GM, temporal lobe, and precuneus in participants aged 8–61 years, and these correlations became significant in additional brain regions, including WM, frontal lobe, ACC, caudate, putamen, amygdala, hippocampus, PHG, and MTL in participants aged 62–92 years. These results suggest that BBB function may be affected by different aspects of the neurovascular function represented by CBF and ATT at different stages of aging.

## BBB kw during normal aging

The kw remained relatively stable throughout early to mid-adulthood, with a marked decline in the early 60 s, especially in males (~18%). This pattern is consistent with previous research on age-related changes in BBB function. For instance, Sweeney and colleagues found evidence for BBB breakdown in the hippocampus, a region crucial of learning and memory, even before clear signs of cognitive dysfunction were evident (*Sweeney et al., 2018*). Similarly, Montagne and colleagues associated early BBB dysfunction with later cognitive impairment, highlighting the vulnerability of the aging brain to vascular changes (*Montagne et al., 2015*). Finally, Taheri and colleagues reported that AQP4 function can differ with age and might contribute to changes in kw (*Taheri et al., 2011*).

We found that specific brain regions are more susceptible to BBB kw decline with aging including regions within each of the four lobes as well as regions that span multiple lobes such as the insula, and cingulate gyrus. BBB dysfunction in this widespread set of these regions could negatively affect a broad range of cognitive processes including executive functions (*Miller and Cohen, 2001*), memory processing (*Eichenbaum and Cohen, 2014*; *Schacter and Wagner, 1999*), emotional regulation (*Craig, 2009*; *Bush et al., 2000*), and sensory perception (*Culham and Kanwisher, 2001*; *Tong, 2003*). Our findings with cognitively normal participants suggest that BBB dysfunction, as indicated by a decline in the kw metric, might precede detectable cognitive impairment. Our results thus suggest that non-invasive imaging techniques such as DP-pCASL may make a significant contribution to future clinical trials seeking to identify and enroll individuals at risk for vascular cognitive impairment for emerging pharmacological treatments.

There is a natural decline and breakdown of BBB function with aging as revealed by both animal and human studies (*Erdő et al., 2017*; *Knox et al., 2022*), which affects all BBB components (endothelial cells, astrocytes, pericytes, microglia, neuronal elements). Other studies reported that the BBB function is relatively stable during early to mid-adulthood but may show alterations with aging (*Zlokovic, 2011*). Several studies have reported that BBB dysfunction and leakage start to increase

in middle-aged individuals and become more pronounced in the elderly, especially in the presence of neurodegenerative diseases (*Sweeney et al., 2019*; *Montagne et al., 2015*; *Tarasoff-Conway et al., 2015*; *Nation et al., 2019*). However, the exact age at which BBB dysfunction begins is still under debate and may vary between individuals. Age-related changes in BBB function can be due to a combination of factors, including reduced expression of tight junction proteins (*Wolburg and Lippoldt, 2002*; *Hawkins et al., 2007*), increased endothelial cell leakage and permeability (*Hawkins and Davis, 2005*), decreased efflux transporter activity (*Cirrito et al., 2005*), and inflammatory changes (*Erickson and Banks, 2018*). An emerging role of AQP4 on BBB function and the glymphatic system has been discovered recently (*Iliff et al., 2012*), and expression and polarization of AQP4 channels can change with aging, which can affect water homeostasis and potentially contribute to age-associated neuropathologies (*Kress et al., 2014*; *Verkman et al., 2014*; *Hubbard et al., 2018*). Our finding of a general trend of kw decline with aging beginning in the early 60 s is consistent with the molecular mechanisms of BBB decline and breakdown during healthy aging.

## Decline in BBB function is more pronounced in elderly males

The most intriguing finding of our study is that BBB kw declines faster in males compared to females in specific brain regions including lateral prefrontal and parietal cortex, as well as lateral and medial temporal lobes. The lateral prefrontal cortex and parietal cortex are heavily interconnected and together support executive functions, planning, and reasoning (*Goldman-Rakic, 1988*). The medial temporal lobe plays a vital role in memory encoding and retrieval (*Squire et al., 2004*). We also observed an asymmetric effect on left and right brain hemispheres, which might be associated with asymmetrically developed vascular burdens in aging (*Giannakopoulos et al., 2009*). Our findings align with evidence indicating that women might maintain certain types of memory skills better than men as they age, suggesting a sex-based resilience in areas of the brain associated with cognitive functions (*McCarrey et al., 2016*; *Sundermann et al., 2016*).

Sex differences in susceptibility to BBB dysfunction, may be influenced by multiple factors including hormonal, genetic, and lifestyle factors (*Arnold and Burgoyne, 2004*). A protective effect of female sex hormones on BBB permeability has been reported, with ovariectomized rats showing increased Evan's blue dye extravasation into the brain which was normalized by estrogen replacement (*Cipolla et al., 2009*; *Wilson et al., 2008*). However, the benefit of estrogen replacement has been observed only in young but not in aged rats (*Deer and Stallone, 2016*). In humans, females show a lower CSF/serum albumin ratio and reduced BBB permeability to Gd-based contrast agents using DCE MRI compared to males (*Parrado-Fernández et al., 2018*; *Moon et al., 2021*). Estrogen is known to have a neuroprotective effect through several mechanisms (*Robison et al., 2019*). The distribution of estrogen receptors on the surface of endothelial cells varies across brain regions, contributing to the observed region-specific kw trajectories between males and females. Additionally, age-related decline in estrogen levels or alterations in estrogen receptors may contribute to the disruption of BBB integrity (*Bake and Sohrabji, 2004*).

Another possibility is that the sex effect we see is related to genetics. Recent studies underscore the X chromosome's integral role in immune function, suggesting its gene expressions may contribute to the sex differences observed in BBB function (*Pinheiro et al., 2011*; *Spolarics, 2007*). Genes, such as tissue inhibitors of matrix metalloproteinase (TIMP), which occasionally escape X-chromosome inactivation, have been linked to protect BBB disruption (*Fujimoto et al., 2008*). Elucidating the genetic underpinnings may illuminate the pathogenesis of BBB-related dysfunctions, with implications for understanding sex differences in neurological disorders.

Additionally, lifestyle factors may contribute to the observed sex differences in BBB integrity. For instance, the prevalence of sleep apnea, which increases with age in men, has been associated with compromised glymphatic clearance (*Lin et al., 2008*). This could partially explain the lower BBB kw observed in older males.

Moreover, understanding the interplay between sex and BBB function might have broader implications. For instance, while estrogen is known to confer some neuroprotective effects (*Brinton, 2009*), this might not be the sole factor responsible for observed sex differences in BBB dysfunction, especially in older age (*Liu et al., 2010*; *Clayton and Collins, 2014*). While the precise mechanisms behind these sex differences in BBB evolution with age remain a topic of ongoing investigation (*Gur and Gur, 2016*), our study and the DP-pCASL technique offer a completely noninvasive approach to probe the

mechanisms underlying these age and sex-related changes in kw, potentially paving the way for early detection and intervention for cognitive decline and neurodegenerative disorders.

## Limitations of the study and future directions

There are a few limitations of this study. A single PLD of 1800 ms was used in this study, which should be sufficient to allow all the labeled water to reach the tissue (i.e. the longest ATT was 1526.7±117.4 and 1468.1±166.9 ms in aged males and females, respectively) (*St Lawrence et al., 2012*). However, a longer PLD should be used in participants with longer expected ATT, such as in stroke and cerebro-vascular disorders. Additionally, a multi-PLD protocol can also be helpful to improve the robustness of quantification accuracy (*Shao et al., 2023*). To compensate for the half signal loss of the non-CPMG DP module, relatively low spatial resolution and TGV-regularized SPA modeling were employed. Our recent development of a motion-compensated diffusion-weighted (MCDW)-pCASL can be utilized to improve the spatial resolution in the future studies (e.g. 3.5 mm$^3$ isotropic maps in 10 min) (*Shao et al., 2023*). Mahroo et al., utilized a multi-echo ASL technique to measure BBB permeability to water and reported shorter intra-voxel transit time and lower BBB exchange time (Tex) in the older participants (≥50 years) compared to the younger group (≤20 years) (*Mahroo et al., 2024*). In animal studies, reduced BBB Tex was also reported in the older mice compared to the younger group using multi-echo ASL (*Ohene et al., 2021*) and a multi-flip-angle, multi-echo dynamic contrast-enhanced (MFAME-DCE) MRI method (*Dickie et al., 2021*). These findings contrast with the results presented in this study, likely due to the different components assessed by different techniques, and increased BBB permeability to water has been suggested to indicate a leakage of tight junctions in aging (*Ohene et al., 2021*; *Dickie et al., 2021*). In contrast, our recent study utilizing high-resolution MCDW-pCASL scans with long averages reveals the potential existence of an intermediate stage of water exchange between vascular and tissue compartments (e.g. paravascular space or basal lamina) (*Shao et al., 2023*). The DP module of the DP-pCASL is hypothesized to null the fast-flowing and pseudo-random oriented spins, which may include both vascular flow and less restricted water in paravascular space. The observed lower kw in older participants may be more related to the delayed exchange across the astrocyte end-feet into the tissue due to loss of the AQP-4 water channel with older age. However, these hypotheses require further investigation to understand the exact mechanisms, especially under different physiological stages (*Zachariou et al., 2024*; *Ying et al., 2024*). Future studies, particularly with animal models targeting specific BBB components under different physiological or diseased conditions, will be valuable for validating these measurements (*Tiwari et al., 2017*; *Ohene et al., 2019*; *Zhang et al., 2019*; *Wei et al., 2023*; *Jia et al., 2023*). Including race as a covariate in our study aims to account for potential variations in brain perfusion observed in previous research (*Leeuwis et al., 2018*; *Clark et al., 2019*). However, it is important to recognize that these differences may not be solely attributable to race. They can be influenced by a complex interplay of factors such as education, environmental exposures, lifestyle, healthcare access, and other social determinants of health (*Williams and Mohammed, 2009*). For example, education has been shown to be highly relevant to regional CBF changes in AD (*Scarmeas et al., 2003*; *Chiu et al., 2004*). Additionally, the potential influence of ancestry and mixed race on perfusion and BBB function requires further investigation in future studies. Other factors such as hematocrit (*Tiwari, 2015*), menopausal status (; *Shao et al., 2020*), and vascular risk factors (*Palomares et al., 2015*) should also be considered. These variables were not included in this study due to the unavailability or limited availability in some cohorts. We attempted to minimize the impact of these factors on our observations by including a relatively large and diverse sample. However, future studies examining the specific mechanism of each of these factors on BBB function in aging would be valuable.

In conclusion, simultaneous mapping of perfusion and BBB kw using DP-pCASL revealed pronounced declines in BBB kw across the lifespan especially in males in their early 60 s, as well as sex-related differences in specific brain regions. Our findings on age and sex-related trajectories of perfusion and BBB variations may provide a foundation for future investigations into perfusion, BBB function, and the glymphatic system in neurodegenerative and other brain disorders.

## Materials and methods

### MRI measurement of BBB kw using DP-pCASL

All participants were scanned on 3T Siemens Prisma scanners using 32 or 64-channel head coils. The imaging parameters of DP-pCASL were: resolution = 3.5×3.5×8 mm$^3$, TR = 4.2 s, TE = 36.2 ms, FOV = 224 mm, 12 slices+10% oversampling, labeling duration = 1.5 s, b=0 and 14 s/mm$^2$ for post-labeling delay (PLD) of 0.9 s with 15 measurements, b=0 and 50 s/mm$^2$ for PLD of 1.8 s with 20 measurements, and total scan time = 10 mins. In addition, 3D MPRAGE for T1 weighted structural MRI (TR = 1.6 s, TI = 0.95 s, TE = 3 ms, 1 mm$^3$ isotropic spatial resolution, 6 min, parameters vary slightly between participating sites) were acquired for segmentation and coregistration.

### Study cohorts

We included 207 cognitively normal participants without current or past major medical illnesses, head injury with cognitive sequelae, intellectual disability, or current substance abuse (MOCA ≥26 and CDR = 0 for elderly participants, middle-aged and young participants were screened for any neurologic or psychiatric disorders or intellectual disability). Participants were recruited from the following cohorts including the University of Southern California (USC) MarkVCID study (https://markvcid.partners.org/) for elderly Latinx participants, the USC Sickle cell trait (SCT) SCT study for African American participants, University of Maryland, the University of Kentucky and USC for racially diverse participants including young to middle-aged to elderly participants. Participants were provided informed consent according to protocols approved by the Institutional Review Board (IRB) of the University of Southern California (USC MarkVCID: HS-19–00662, USC SCT: HS-18–00826), University of Maryland (HP-00082052) and University of Kentucky (IRB number 46325). Among recruited participants, 21 of them were excluded due to a lack of demographic information or severe image artifacts. A total of 186 participants (97 females and 89 males) with ages ranging from 8 to 92 years old were included for further analysis, including 65 white, 27 Latinx, 47 African American, and 47 Asian participants. Detailed distributions of age, sex, and race are shown in Supplemental Fig. S1. All participants were asked to avoid caffeine at least 3 hr ahead of the scan and stay awake during the DP-pCASL scan. Additional head padding was used to reduce head movement during the scan. For more vulnerable participants (participants under 17 years old), we had instructed them to stare at the cross shown on the screen behind the scanner to minimize head motion.

### Image preprocessing

DP-pCASL data was analyzed using our in-house-developed LOFT BBB toolbox including motion correction, physiological noise reduction using principal component analysis (PCA) (*Shao et al., 2017*), and calculation of signal differences between the labeled and control images. ATT maps were generated using DP-pCASL signals acquired at the PLD = 0.9 s with b=0, 14 s/mm$^2$ (FEAST method) (*Wang et al., 2003*), and CBF calculated using the PLD = 1.8 s data according to the single-compartment Buxton model (*Buxton et al., 1998*). The kw map was generated with a total-generalized-variation (TGV)-regularized single-pass approximation (SPA) model using DP-pCASL signals acquired at the PLD = 1.8 s with b=0, 50 s/mm$^2$, as well as CBF, ATT, and tissue relaxation time (R$_{1b}$) map generated from background suppressed control images (*Shao et al., 2019*). Intracranial volume (ICV) and voxel-wise gray matter (GM) density was obtained by segmentation of T1w MPRAGE images using SPM12 (http://fil.ion.ucl.ac.uk/spm/). CBF, ATT, and kw maps were co-registered to the T1w MPRAGE images along with the M0 image of DP-pCASL, and then normalized into the Montreal Neurological Institute (MNI) standard brain template (2 mm$^3$ isotropic resolution) using SPM12 before group-level analysis.

### Regional analysis using MARS

Anticipating that associations of regional CBF, ATT, kw with age may be nonlinear, especially if BBB degradation initiates within a specific age range, we employed MARS (*Friedman, 1991*) for analysis, which is adept at automatically identifying nonlinearities (splines) and interactions among variables. To mitigate the risk of overfitting, we conducted a 10-fold generalized cross-validation (GCV), aiming to minimize outlier influence and reduce model complexity. This approach ensures an optimal prediction model boasting the best generalizability. The final MARS model, utilizing the hockey stick basis function, yielded predicted values for the testing sample, with each prediction grounded in models trained on mutually exclusive data sets following the 10-fold cross-validation. The final model performance

was presented as the R$^2$ between model-predicted and the observed value. The predicted spline slope with a corresponding 95% confidence interval for the expected value at each age point was illustrated using a scatter plot with a spline trajectory curve. We have fit the MARS model for each imaging feature (CBF, ATT, kw) in 14 subregions including GM, WM, frontal lobe, temporal lobe, parietal lobe, anterior cingulate cortex (ACC), posterior cingulate cortex (PCC), precuneus, caudate, amygdala, hippocampus, parahippocampal gyrus (PHG) and medial temporal lobe (MTL) based on the Automated Anatomical Atlas (AAL) template (*Tzourio-Mazoyer et al., 2002*). Unlike the conventional statistical modeling, MARS is a machine learning tool with the output mainly focuses on the predicted accuracy from cross-validation. To facilitate the model interpretation, the basis functions selected by MARS as the important predictors were refitted in generalized linear model to obtain the slope estimation with a 95% confidence interval. SAS 9.4 was used for MARS model fitting and validation.

### Voxel-wise analysis using GLM

We employed two GLM models to study the voxel-wise effect of age and the interaction effects between age and sex on CBF, ATT, and kw. The models are detailed as follows:

### Age effect analysis (GLM1)

This model was designed to investigate the influence of age on the dependent variables. The equation for GLM1 is as follows:

$$Y \sim \text{intercept} + \text{age} + \text{covariates(sex} + \text{race} + \text{ICV} + \text{GM density} + Z) \tag{1}$$

### Interaction effect analysis of age and sex (GLM2)

GLM2 aimed to analyze the interaction effects between age and sex on the dependent variables. The equation for GLM2 is expressed as:

$$Y \sim \text{intercept} + \text{age} + \text{sex} + \text{age} \times \text{covariates(race} + \text{ICV} + \text{GM density} + Z)$$

In both models, 'Y' represents an individual imaging feature (either CBF, ATT, or kw), while the remaining two features are included as covariates (represented as 'Z' in equations [1, 2]). For instance, when analyzing kw changes, 'Y' denotes kw, and 'Z' includes CBF and ATT.

For the statistical analysis, we corrected for multiple comparisons using AlphaSim (*Cox, 1996*). We identified clusters comprising more than 501 voxels as significant, with a global α level of 0.05.

## Acknowledgements

This work was supported by the National Institute of Health (NIH) grant R01-NS134712, UH3-NS100614, S10-OD032285, R01-NS114382, R01-EB028297, R21-EY028721, RF1-NS122028, R01-AG068055, P30- AG072946, R01-MH116948, and RF1-NS114628.

## Additional information

### Funding

| Funder | Grant reference number | Author |
|---|---|---|
| National Institutes of Health | UH3-NS100614 | Danny JJ Wang |
| National Institutes of Health | S10-OD025312 | Danny JJ Wang |
| National Institutes of Health | R01-NS114382 | Danny JJ Wang |
| National Institutes of Health | R01-EB028297 | Danny JJ Wang |
| National Institutes of Health | R21-EY028721 | Xuejuan Jiang |

| Funder | Grant reference number | Author |
|--------|------------------------|--------|
| National Institutes of Health | RF1-NS122028 | Brian T Gold<br>Danny JJ Wang |
| National Institutes of Health | R01-AG068055 | Brian T Gold |
| National Institutes of Health | P30- AG072946 | Brian T Gold |
| National Institutes of Health | R01-MH116948 | Elliot Hong |
| National Institutes of Health | RF1-NS114628 | Peter Kochunov<br>Elliot Hong |
| National Institutes of Health | R01-NS134712 | Xingfeng Shao<br>Danny JJ Wang |

The funders had no role in study design, data collection and interpretation, or the decision to submit the work for publication.

## Author contributions

Xingfeng Shao, Conceptualization, Data curation, Software, Formal analysis, Funding acquisition, Validation, Investigation, Visualization, Methodology, Writing – original draft, Project administration, Writing – review and editing; Qinyang Shou, Data curation, Formal analysis, Writing – review and editing; Kimberly Felix, Resources, Data curation; Brandon Ojogho, Kay Jann, Resources, Data curation, Formal analysis, Writing – review and editing; Xuejuan Jiang, Brian T Gold, Megan M Herting, Eric L Goldwaser, Peter Kochunov, Elliot Hong, Resources, Data curation, Writing – review and editing; Ioannis Pappas, Meredith Braskie, Resources, Data curation, Investigation, Writing – review and editing; Hosung Kim, Resources, Data curation, Formal analysis, Investigation, Visualization, Writing – review and editing; Steven Cen, Supervision, Validation, Investigation, Methodology, Writing – original draft, Writing – review and editing; Danny JJ Wang, Conceptualization, Resources, Data curation, Supervision, Funding acquisition, Writing – review and editing

## Author ORCIDs

Xingfeng Shao ⓘ https://orcid.org/0000-0002-4130-6204
Danny JJ Wang ⓘ http://orcid.org/0000-0002-0840-7062

## Ethics

Participants were provided informed consent according to protocols approved by the Institutional Review Board (IRB) of the University of Southern California (USC MarkVCID: HS-19-00662, USC SCT: HS-18-00826), University of Maryland (HP-00082052) and University of Kentucky (IRB number 46325).

Reviewer #1 (Public review): https://doi.org/10.7554/eLife.96155.3.sa1
Reviewer #2 (Public review): https://doi.org/10.7554/eLife.96155.3.sa2
Author response https://doi.org/10.7554/eLife.96155.3.sa3

# Additional files

## Supplementary files

• Supplementary file 1. Supplementary tables. (**a**) Analysis of age and sex-dependent kw variations in 14 brain regions using MARS. Threshold ages when kw starts declining have been identified. The age ×sex interaction terms being negative suggest a more pronounced decline in kw in males compared to females. For females, the post-threshold age kw decline slope is presented with 95% confidence intervals (CI) and P values. The male kw decline slopes were estimated from the female slope and the age ×sex interaction. Entries labeled 'N.A. (N.S.)' indicate non-significant changes. (**b**) Analysis of age and sex dependent CBF variations in 14 brain regions using MARS. Threshold ages where CBF slope changes occur were identified. A negative age ×sex interaction term was detected only in the hippocampus, suggesting a more pronounced CBF decline in males compared to females. In other brain regions, the rate of CBF decline with age is largely similar between males

and females, although males consistently exhibit lower CBF. For both males and females, the CBF decline slopes before and after the threshold age are presented with 95% confidence intervals (CIs) and P values. (**c**) Analysis of age and sex dependent ATT variations in 14 brain regions using MARS. Threshold ages where ATT slope changes occur were identified. The rate of ATT increase with age is largely similar between males and females, although males consistently exhibit longer ATT. For both males and females, the ATT increase slopes before and after the threshold age are presented with 95% confidence intervals (CIs) and P values. (**d**) Voxel-wise analysis of age trend in kw. Brain regions with significant negative correlations between kw and age with corresponding location (AAL template), cluster size, T values, and P values. (**e**) Voxel-wise analysis of age and sex effect in kw, CBF, and ATT. Brain regions with significant age ×sex effects in kw, CBF, and ATT with corresponding locations (AAL template), cluster sizes, T values, and P values. (**f**) Association between kw and CBF or ATT in 14 brain regions for participants aged between 8–61 years and 62–92 years. Linear regressions incorporating sex as a covariate were conducted, and the estimated coefficients were presented with 95% confidence intervals (CI) and P values.

- MDAR checklist

### Data availability

The reconstruction toolbox presented in this manuscript can be found on our LOFT lab website (https://loft-lab.org/index-5.html under tab 'Water exchange rate (kw) across the blood-brain barrier quantification toolbox'). To request this toolbox, please send an email with your information (also Mac or windows version, Linux not supported currently) to Dr. Danny JJ Wang (jj.wang@loni.usc.edu). The reconstruction toolbox will be available after we establish Material Transfer Agreement (MTA) between user's institute and University of Southern California. NIFTI files for CBF, ATT and kw maps along with demographic information has been uploaded to OpenNeuro (https://openneuro.org/datasets/ds005529).

The following dataset was generated:

| Author(s) | Year | Dataset title | Dataset URL | Database and Identifier |
|---|---|---|---|---|
| Shao X, Shou Q, Felix K, Ojogho B, Jiang X, Gold BT, Herting MM, Goldwaser EL, Kochunov P, Hong LE, Pappas I, Braskie M, Kim H, Cen S, Jann K, Wang DJJ | 2024 | DP-pCASL data (CBF, ATT, BBB kw) from 186 cognitively normal participants (8-92 years) | https://openneuro.org/datasets/ds005529/ | OpenNeuro, ds005529 |

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
