## [Editor Report · eLife assessment]

This study presents a **valuable** finding that the blood-brain barrier functionality changes with age and differs between males and females. The analysis is **solid**, comprising a large and racially diverse dataset, and utilizes a contrast-agent-free MRI method. Since limited work has been done in the MRI field on the blood-brain barrier using this method, this study is of great interest to neuroimaging researchers and clinicians.

---

## [Referee Report · Reviewer #1 (Public review)]

Summary:

This work revealed an important finding that the blood-brain barrier (BBB) functionality changes with age and is more pronounced in males. The authors applied a non-invasive, contrast-agent-free approach of MRI called diffusion-prepared arterial spin labeling (DP-pCASL) to a large cohort of healthy human volunteers. DP-pCASL works by tracking the movement of magnetically labeled water (spins) in blood as it perfuses brain tissue. It probes the molecular diffusion of water, which is sensitive to microstructural barriers, and characterizes the signal coming from fast-moving spins as blood and slow-moving spins as tissue, using different diffusion gradients (b-values). This differentiation is then used to assess the water exchange rates (kw) across the BBB, which acts as a marker for BBB functionality. The main finding of the authors is that kw decreases with age, and in some brain regions, kw decreased faster in males. The neuroprotective role of the female sex hormone, estrogen, on BBB function is discussed as one of the explanations for this finding, supported by literature. The study also shows that BBB function remains stable until the early 60s and remarkably decreases thereafter.

Strengths:

The two main strengths of the study are the MRI method used and the amount of data. The authors employed a contrast-agent-free MRI method called ASL, which offers the opportunity to repeat such experiments multiple times without any health risk-a significant advantage of ASL. Since ASL is an emerging field that requires further exploration and testing, a study evaluating blood-brain barrier functionality is of great importance. The authors utilized a large dataset of healthy humans, where volunteer data from various studies were combined to create a substantial pool. This strategy is effective for statistically evaluating differences in age and gender.

Weaknesses:

The findings are of great interest as this assessment is the first of its kind to assess BBB function using ASL. Further studies are needed to compare DP-ASL findings with more established methods, such as PET and BBB molecular/ blood biomarkers.

---

## [Referee Report · Reviewer #2 (Public review)]

Summary:

This study used a novel diffusion-weighted pseudo-continuous arterial spin labelling (pCASL) technique to simultaneously explore age- and sex-related differences in brain tissue perfusion (i.e., cerebral blood flow (CBF) & arterial transit time (ATT) - a measure of CBF delivery to brain tissue) and blood-brain barrier (BBB) function, measured as the water exchange (kw) across the BBB. While age- and sex-related effects on CBF are well known, this study provides new insights to support the growing evidence of these important factors in cerebrovascular health, particularly in BBB function. Across the brain, decline in CBF and BBB function (kw) and elevation in ATT was reported in older adults, after the age of 60 and more so in males compared to females. This was also evident in key cognitive regions including the insular, prefrontal, and medial temporal regions, stressing the consideration of age and sex in these brain physiological assessments.

Strengths:

Simultaneous assessment of CBF with BBB along with transit time and at the voxel-level helped elucidate the brain's vulnerability to age and sex-effects. It is apparent that the investigators carefully designed this study to assess regional associations of age and sex with attention to exploring potential non-linear effects.

Weaknesses:

It appears that no brain region showed concurrent CBF and BBB dysfunction (kw), based on the results reported in the main manuscript and supplemental information. Was an association analysis between CBF and kw performed? There is a potential effect of the level of formal education on CBF (PMID: 12633147; 15534055), which could have been considered and accounted for as well, especially for a cohort with stated diversity (age, race, sex).

---

## [Author Response]

The following is the authors’ response to the original reviews.

**Reviewer #1 (Public Review):**
Summary:This work revealed an important finding that the blood-brain barrier (BBB) functionality changes with age and is more pronounced in males. The authors applied a non-invasive, contrast-agent-free approach of MRI called diffusion-prepared arterial spin labeling (DP-pCASL) to a large cohort of healthy human volunteers. DP-pCASL works by tracking the movement of magnetically labeled water (spins) in blood as it perfuses brain tissue. It probes the molecular diffusion of water, which is sensitive to microstructural barriers, and characterizes the signal coming from fast-moving spins as blood and slow-moving spins as tissue, using different diffusion gradients (b-values). This differentiation is then used to assess the water exchange rates (kw) across the BBB, which acts as a marker for BBB functionality. The main finding of the authors is that kw decreases with age, and in some brain regions, kw decreases faster in males. The neuroprotective role of the female sex hormone, estrogen, on BBB function is discussed as one of the explanations for this finding, supported by literature. The study also shows that BBB function remains stable until the early 60s and remarkably decreases thereafter.Strengths:The two main strengths of the study are the MRI method used and the amount of data. The authors employed a contrast-agent-free MRI method called ASL, which offers the opportunity to repeat such experiments multiple times without any health risk - a significant advantage of ASL. Since ASL is an emerging field that requires further exploration and testing, a study evaluating blood-brain barrier functionality is of great importance. The authors utilized a large dataset of healthy humans, where volunteer data from various studies were combined to create a substantial pool. This strategy is effective for statistically evaluating differences in age and gender.Weaknesses:R1.0: Gender-related differences are only present in some brain regions, not in the whole brain or gray matter - which is usually the assumption unless stated otherwise. From the title, this was not clear. Including simulations could increase readers' understanding related to model fitting and the interdependence of parameters, if present. The discussion follows a clear line of argument supported by literature; however, focusing solely on AQP4 channels and missing a critical consideration of other known/proven changes in transport mechanisms through the BBB and their effects substantially weakens the discussion.

Thanks for your insightful feedback and suggestions. We have made the following changes to the manuscript:

(1) The title has been modified to highlight the sex differences in specific brain regions: “Age-Related Decline in Blood-Brain Barrier Function is More Pronounced in Males than Females in Parietal and Temporal Regions.”

(2) To study the potential impact of prolonged ATT seen in males on estimated kw, we simulated kw distribution for females by adjusting ATT by +60 ms to match males' ATT. This led to marginally higher kw values (Supplemental Figure S2), suggesting that the kw difference between males and females is not a direct result of prolonged ATT. Additionally, we have added a section titled “Data and Code Availability Statements” in the revised manuscript to indicate that we are willing to share the reconstruction toolbox with interested groups. The toolbox is a standalone MATLAB-based program (no license required) to generate kw, CBF, and ATT maps, which can run on Windows or Mac computers.

(3) We agree with the reviewer that BBB water exchange can be facilitated by other transport mechanisms, as we mentioned in the introduction: “Water exchange across the BBB occurs at a relatively high level and is mediated by passive diffusion, active co-transport through the endothelial membrane, and facilitated diffusion through the dedicated water channel, aquaporin-4 (AQP4), at the end-feet of astrocytes.” We emphasized our findings related to AQP4 based on the technical properties of DP-pCASL, which is more sensitive to the exchange occurring across astrocyte end-feet. We also acknowledge that different techniques can be helpful to study other components of BBB water exchange, and we have added the following discussion to the updated manuscript: “Mahroo et al., utilized a multi-echo ASL technique to measure BBB permeability to water and reported shorter intra-voxel transit time and lower BBB exchange time (Tex) in the older participants (≥50 years) compared to the younger group (≤20 years). In animal studies, reduced BBB Tex was also reported in the older mice compared to the younger group using multi-echo ASL and a multi-flip-angle, multi-echo dynamic contrast-enhanced (MFAME-DCE) MRI method. These findings contrast with the results presented in this study, likely due to the different components assessed by different techniques, and increased BBB permeability to water has been suggested to indicate a leakage of tight junctions in aging. In contrast, our recent study utilizing high resolution MCDW-pCASL scans with long averages reveals the potential existence of an intermediate stage of water exchange between vascular and tissue compartments (e.g., paravascular space or basal lamina). The DP module of the DP-pCASL is hypothesized to null the fast-flowing and pseudo-random oriented spins, which may include both vascular flow and less restricted water in paravascular space. The observed lower kw in older participants may be more related to the delayed exchange across the astrocyte end-feet into the tissue due to loss of AQP-4 water channel with older age. However, these hypotheses require further investigation to understand the exact mechanisms, especially under different physiological states. Future studies, particularly with animal models targeting specific BBB components under different physiological or diseased conditions, will be valuable for validating these measurements.”

**Reviewer #1 (Recommendations For The Authors):**
R1.1 The manuscript is well-organized and presents arguments in a logical order. The visual representation of results in the form of figures is sufficient (see style suggestions below).

Thanks for your suggestions on improving the figures, we have updated figures for better visualization (Please see our response to R1.5, R1.6, R1.7 and R1.8).

R1.2 It would be beneficial if the model/toolbox could be made publicly available so that fellow researchers from the community could apply and test it in their research.

We have added a section “Data and code availability statements” in the revised manuscript to indicate we’re willing to share the toolbox to the interested groups (L529 in the annotated manuscript). The toolbox is a standalone MATLAB-based program (no license required) to generate kw, CBF and ATT maps, which can run on windows or MAC computers. Indeed, we have been sharing our reconstruction toolbox with over 50 collaboration sites. The following screenshots are examples of three steps performed by the toolbox (shared by one collaborator):

**Author response image 1. sa3fig1:** Step 1: Loading raw data and calculate T1 map.

**Author response image 2. sa3fig2:** Step 2: Motion correction and skull stripping.

**Author response image 3. sa3fig3:** Step 3: kw, CBF and ATT quantification (nii files will be saved).

R1.3 Line 46 states that the technique is novel, but it has been introduced and used before (Shao, et al. MRM 2019). It sure is innovative but the term novel is too strong and may confuse the readers that it is something new introduced in this manuscript.

Thanks for the suggestion, we agree the term ‘novel’ may cause confusion about the technique, we have removed it in the revised manuscript (L48, L50).

R1.4 Line 395, kw was generated using PLD = 1.8s with b = 0, 50 s/mm2. Is only one-time point enough for estimating kw? To me, it is not clear how robust is the kw estimation with only one PLD.

According to the single-pass approximation (SPA) model (1), kw can be accurately estimated when the PLD is longer than the ATT. We recruited cognitively normal participants in this study and found the longest ATT to be 1526.7±117.4 and 1468.1±166.9 ms in aged (62-92 years) males and females, respectively. A PLD of 1.8 s was chosen to balance the SNR of the data and the accuracy of the model fitting, which should be sufficient for this study. However, for future studies involving diseased populations with prolonged ATT, a longer PLD should be used, or a multi-PLD protocol could be helpful to improve the robustness of quantification accuracy.

We have added a limitation statement in the revised manuscript (L407): "A single PLD of 1800 ms was used in this study, which should be sufficient to allow all the labeled water to reach the tissue (i.e., the longest ATT was 1526.7±117.4 and 1468.1±166.9 ms in aged males and females, respectively) (1). However, a longer PLD should be used in participants with longer expected ATT, such as in stroke and cerebrovascular disorders. Additionally, a multi-PLD protocol can also be helpful to improve the robustness of quantification accuracy (2)."

R1.5 Suggestion: Figure 3A, colormap for kw appears suboptimal. Regional differences are hard to see.

Thanks for the suggestion, we have updated the range of color scale (from [0, 200], to [70, 160]) to highlight the regional differences in the updated Figure 3:

We prefer to use the same blue colormap that we and our collaborators have been using this for publications to maintain consistence. We also acknowledged the limitation of the spatial resolution of kw maps in the updated manuscript (L412): “To compensate for the half signal loss of the non-CPMG DP module, relatively low spatial resolution and TGV-regularized SPA modeling were employed. Our recently development of a motion-compensated diffusion weighted (MCDW)-pCASL can be utilized to improve the spatial resolution in the future studies (e.g. 3.5 mm3 isotropic maps in 10 mins) (2)”

R1.6 Suggestion: use same/similar colormaps for the same parameters (kw, ATT, CBF) to help the reader follow across Figures 3, 4, and 5.

Thanks for your suggestion, we agree that using the same color would be easier for readers to follow the context. However, figures 4 and 5 were created to show the age and sex dependent changes, so that we used warm and cold colors to indicate effects of decrease and increase, respectively. We clarified the choice of colormap in the figure captions (L260, L284): “The effects of decrease or increase were represented by warm colors (yellow to red) and cold (gray to blue) colors, respectively.”

R1.7 Suggestion: please be consistent with the ordering of parameters in Figures 3, 4, and 5.

Thanks for the suggestion, we have updated Figure 3 to consistently show kw, CBF and ATT results in order from left to right:

R1.8 Suggestion: use the same scaling (e.g.[|1.9|, |11 |] for Fig. 4, [|1.9|, |4|] for Figure 5) to enhance comparability across parameters in the subfigures.

Thanks for the suggestion, we agree that the same scaling would enhance the comparability across parameters. We have updated the color scales for Figure 5 using maximal |T| = 4:

However, range of maximal |T| was relatively large for Figure 4 (i.e. 5 for kw, 11 for CBF and 7 for ATT), and using the same color scale might oversaturate the regional responses or diminish the visibility of regional differences. Therefore, we prefer to keep the original color scale for Figure 4.

R1.9 In Figure 5, the interaction of age with sex in kw parameter seems to be more on one side of the brain. What could be the reasons for possible lateralization?

We agree with the reviewer that the age and sex interaction effects emphasized on one side is an interesting finding. While we do not have a clear explanation now, we suspect it may relate to aging-related asymmetrical vascular burdens. Giannakopoulos et al. reported that vascular scores, indicating higher vascular burden, were significantly higher in the left hemisphere across all Clinical Dementia Rating scores. Moreover, the predominance of Alzheimer’s disease and vascular pathology in the right hemisphere correlated with significantly higher Clinical Dementia Rating scores (3). We added the following to the updated manuscript to discuss this potential mechanism (L370): “… We also observed an asymmetric effect on left and right brain hemispheres, which might be associated with asymmetrically developed vascular burdens in aging (3)."

R1.10 A comparison between the present study and DCE MRI as well as other ASL methods evaluating BBB function with age is missing. ASL techniques probing transverse relaxation and DCE MRI have reported increased kw with age in humans as well as in animal models. What could be the reasons?

We agree with the reviewer that BBB water exchange measured by other methods should be sufficiently discussed, especially regarding their age-related changes. We added the following discussion in the updated manuscript (L415): “Mahroo et al., utilized a multi-echo ASL technique to measure BBB permeability to water and reported shorter intra-voxel transit time and lower BBB exchange time (Tex) in the older participants (≥50 years) compared to the younger group (≤20 years) (4). In animal studies, reduced BBB Tex was also reported in the older mice compared to the younger group using multi-echo ASL (5) and a multi-flip-angle, multi-echo dynamic contrast-enhanced (MFAME-DCE) MRI method (6). These findings contrast with the results presented in this study, likely due to the different components assessed by different techniques, and increased BBB permeability to water has been suggested to indicate a leakage of tight junctions in aging (5, 6). In contrast, our recent study utilizing high resolution MCDW-pCASL scans with long averages reveals the potential existence of an intermediate stage of water exchange between vascular and tissue compartments (e.g., paravascular space or basal lamina) (2). The DP module of the DP-pCASL is hypothesized to null the fast-flowing and pseudo-random oriented spins, which may include both vascular flow and less restricted water in paravascular space. The observed lower kw in older participants may be more related to the delayed exchange across the astrocyte end-feet into the tissue due to loss of AQP-4 water channel with older age. However, these hypotheses require further investigation to understand the exact mechanisms, especially under different physiological states (7, 8). Future studies, particularly with animal models targeting specific BBB components under different physiological or diseased conditions, will be valuable for validating these measurements (9-13).”

R1.11 Line 163/164, a rapid decrease of CBF in males in the region of the hippocampus is reported. It would be beneficial to discuss this in discussion further (has this been reported before, possible reasons, etc).

Thanks for the suggestion, we agree that the accelerated CBF decline in males in the hippocampus is an important finding, we have added discussion in the revised manuscript (L300): "Furthermore, we found a more pronounced age-related decline in CBF in the hippocampus of males compared to females (Fig. 2, Supplemental Table S2). To the best of our knowledge, no study has previously reported this accelerated hippocampal CBF decline in males. This finding may be linked to the accelerated hippocampal volume loss in males, as reported in a study analyzing 19,793 generally healthy UK Biobank participants (14). Lower hippocampal perfusion has been associated with poor memory performance (15, 16), suggesting that males might be more vulnerable to potential cognitive decline (17).

R1.12 Lines 198-202 describe a simulation done to test the dependence of kw on ATT. This is important and could be explained more in detail. Adding simulation results (numeric or figure) to supplementary materials would increase reproducibility and understanding for others.

We apologize for not referencing to the simulation results in the main text. We simulated kw distribution for females by adjusting ATT by +60 ms to matching males’ ATT, leading to a marginally higher kw values. And these results were shown in the Supplemental Figure S2 C (yellow):

We have now referenced the simulation results in the updated manuscript (L206).

R1.13 No limitations of the presented work are mentioned. A critical perspective would increase the scientific impact on future research decisions and implementation of this method by others.

Thanks for the suggestion, we agree the limitations need to be acknowledged. We have added a limitation paragraph in the revised manuscript (L406): "Limitations of the study and future directions: There are a few limitations of this study. A single PLD of 1800 ms was used in this study, which should be sufficient to allow all the labeled water to reach the tissue (i.e., the longest ATT was 1526.7±117.4 and 1468.1±166.9 ms in aged males and females, respectively) (1). However, a longer PLD should be used in participants with longer expected ATT, such as in stroke and cerebrovascular disorders. Additionally, a multi-PLD protocol can also be helpful to improve the robustness of quantification accuracy (2). To compensate for the half signal loss of the non-CPMG DP module, relatively low spatial resolution and TGV-regularized SPA modeling were employed. Our recently development of a motion-compensated diffusion weighted (MCDW)-pCASL can be utilized to improve the spatial resolution in the future studies (e.g. 3.5 mm3 isotropic maps in 10 mins) (2). Mahroo et al., utilized a multi-echo ASL technique to measure BBB permeability to water and reported shorter intra-voxel transit time and lower BBB exchange time (Tex) in the older participants (≥50 years) compared to the younger group (≤20 years) (4). In animal studies, reduced BBB Tex was also reported in the older mice compared to the younger group using multi-echo ASL (5) and a multi-flip-angle, multi-echo dynamic contrast-enhanced (MFAME-DCE) MRI method (6). These findings contrast with the results presented in this study, likely due to the different components assessed by different techniques, and increased BBB permeability to water has been suggested to indicate a leakage of tight junctions in aging (5, 6). In contrast, our recent study utilizing high resolution MCDW-pCASL scans with long averages reveals the potential existence of an intermediate stage of water exchange between vascular and tissue compartments (e.g., paravascular space or basal lamina) (2). The DP module of the DP-pCASL is hypothesized to null the fast-flowing and pseudo-random oriented spins, which may include both vascular flow and less restricted water in paravascular space. The observed lower kw in older participants may be more related to the delayed exchange across the astrocyte end-feet into the tissue due to loss of AQP-4 water channel with older age. However, these hypotheses require further investigation to understand the exact mechanisms, especially under different physiological stages (7, 8). Future studies, particularly with animal models targeting specific BBB components under different physiological or diseased conditions, will be valuable for validating these measurements (9-13). Including race as a covariate in our study aims to account for potential variations in brain perfusion observed in previous research (18, 19). However, it is important to recognize that these differences may not be solely attributable to race. They can be influenced by a complex interplay of factors such as education, environmental exposures, lifestyle, healthcare access, and other social determinants of health (20). For example, education has been shown to be highly relevant to regional CBF changes in AD (21, 22). Additionally, the potential influence of ancestry and mixed-race on perfusion and BBB function requires further investigation in future studies. Other factors such as hematocrit (23), menopausal status (24, 25), and vascular risk factors (26) should also be considered. These variables were not included in this study due to the unavailability or limited availability in some cohorts. We attempted to minimize the impact of these factors on our observations by including a relatively large and diverse sample. However, future studies examining the specific mechanism of each of these factors on BBB function in aging would be valuable.

**Reviewer #2 (Public Review):**
Summary:This study used a novel diffusion-weighted pseudo-continuous arterial spin labelling (pCASL) technique to simultaneously explore age- and sex-related differences in brain tissue perfusion (i.e., cerebral blood flow (CBF) & arterial transit time (ATT) - a measure of CBF delivery to brain tissue) and blood-brain barrier (BBB) function, measured as the water exchange (kw) across the BBB. While age- and sex-related effects on CBF are well known, this study provides new insights to support the growing evidence of these important factors in cerebrovascular health, particularly in BBB function. Across the brain, the decline in CBF and BBB function (kw) and elevation in ATT were reported in older adults, after the age of 60, and more so in males compared to females. This was also evident in key cognitive regions including the insular, prefrontal, and medial temporal regions, stressing the consideration of age and sex in these brain physiological assessments.Strengths:Simultaneous assessment of CBF with BBB along with transit time and at the voxel-level helped elucidate the brain's vulnerability to age and sex-effects. It is apparent that the investigators carefully designed this study to assess regional associations of age and sex with attention to exploring potential non-linear effects.Weaknesses:R2.0 It appears that no brain region showed concurrent CBF and BBB dysfunction (kw), based on the results reported in the main manuscript and supplemental information. Was an association analysis between CBF and kw performed? There is a potential effect of the level of formal education on CBF (PMID: 12633147; 15534055), which could have been considered and accounted for as well, especially for a cohort with stated diversity (age, race, sex).

Thank you for your positive feedback and comments on the potential associations between BBB kw and other physiological parameters (e.g., CBF) and socioeconomic factors (e.g., education). We have made the following changes to the updated manuscript:

(1) We conducted additional linear regressions between regional kw and regional CBF or ATT, incorporating sex as a covariate, for participants aged 8-61 years and 62-92 years (when BBB kw starts declining). The results are summarized in Supplemental Table S6. We found that BBB kw was significantly negatively associated with CBF in the putamen, amygdala, hippocampus, parahippocampal gyrus, and medial temporal lobe in participants younger than 62 years, when kw was relatively consistent across ages. However, no significant correlations were found in any brain regions in the 62-92 years group. In contrast to CBF, kw was significantly negatively associated with ATT in the GM, temporal lobe, and precuneus in participants aged 8-61 years, and these correlations became significant in additional ROIs, including WM, frontal lobe, ACC, caudate, putamen, amygdala, hippocampus, PHG, and MTL in participants aged 62-92 years. These results suggest that BBB function may be influenced by different aspects of neurovascular function represented by CBF and ATT at different stages of aging.

(2) One limitation of this study is the lack of information on participants’ geographical, cultural, physical characteristics, and socioeconomic factors. While we included race as a covariate to account for potential variations observed in previous research, race is an imprecise proxy for the complex interplay of genetic, environmental, socioeconomic, and cultural factors that influence physiological outcomes. We have acknowledged this limitation by adding the following discussion in the updated manuscript: “Including race as a covariate in our study aims to account for potential variations in brain perfusion observed in previous research. However, it is important to recognize that these differences may not be solely attributable to race. They can be influenced by a complex interplay of factors such as education, environmental exposures, lifestyle, healthcare access, and other social determinants of health. For example, education has been shown to be highly relevant to regional CBF changes in AD. Additionally, the potential influence of ancestry and mixed-race on perfusion and BBB function requires further investigation in future studies.”

**Reviewer #2 (Recommendations For The Authors):**
General comments:I commend the authors on a very well-written and laid-out study. General remarks have been provided in the short assessment and public review sections.

We would like to thank the reviewer for the insightful suggestions and overall positive feedback. We have substantial revised and improved our manuscript, and point-to-point responses can be found in the following sections and in the annotated manuscript.

Specific comments:Results:R2.1 Line 127: "since race may influence the changes in perfusion and kw with aging, it was included as a covariate". It is not clear how race - a simplistic term for ethnicity or to be more specific ancestry has been shown to influence changes in perfusion? Is it known for a fact that for example, older Black people have lower/higher CBF or kw compared to Asians or Asians to Caucasian Americans? Can this be extrapolated to Japanese Brazilians having different patterns of regional CBF to Caucasian or Black Brazilians or similar patterns of CBF to Japanese people in Japan since they share similar race? Do Dutch people in the Netherlands share CBF characteristics to their descendants in the US or in South Africa? Would the geographical, cultural, and other physical characteristics of one's ethnicity or lineage impact CBF? Race is often used as a poor substitute for the complex interactions of physical, socioeconomic, and geopolitical factors that produce disparities that may have measurable biological effects including CBF. But it is not clear why being one race vs the other will impact CBF, without carefully parcelling out the many factors beyond biology, if any. Is any of the participants in the study mixed race? How about recently settled individuals who may identify for example as Black but have spent all their life up to adult years outside of the US and marked here in the study as simply African American? Not that I am saying this is the case. However this simplification may require more careful analysis.

In our study, no participant indicated to be mixed-race, and unfortunately we do not have additional information about their specific ancestry or information about their geographical, cultural, and other physical characteristics. We acknowledge that race is an imprecise proxy for the complex interplay of genetic, environmental, socioeconomic, and cultural factors that influence physiological outcomes, including perfusion and BBB function. The use of race as a covariate in our study is intended to account for potential variations observed in previous research, rather than to imply a direct causal relationship.

Research has shown differences in blood flow among racial groups (18, 19). However, these differences are not solely attributable to race, and they are also shaped by environmental exposures, lifestyle factors, healthcare access, and other social determinants of health (20). We have added the following discussion in the updated manuscript (L436): “Including race as a covariate in our study aims to account for potential variations in brain perfusion observed in previous research (18, 19). However, it is important to recognize that these differences may not be solely attributable to race. They can be influenced by a complex interplay of factors such as education, environmental exposures, lifestyle, healthcare access, and other social determinants of health (20). For example, education has been shown to be highly relevant to regional CBF changes in AD (21, 22). Additionally, the potential influence of ancestry and mixed-race on perfusion and BBB function requires further investigation in future studies.”

R2.2 Figure 3: Could the standard deviation of the reported values be also stated so the variance can be appreciated?

Thanks for the suggestion, we have added the standard deviation of the kw, CBF and ATT values on the updated Figure 3:

R2.3 Discussions: Line 280: .."observed distinct trajectory of kw changes with aging as compared with CBF and ATT. I presume this as compared to the earlier statements (line 268) of pervasive increase in ATT and decrease in CBF across the brain. Were there any brain regions that showed increased ATT, decreased CBF and kw as a function of age or even sex?? Was there any association between CBF and kw in any brain regions, across the participants after controlling for sex differences? If there is a suspicion of early BBB dysfunction (line 286) preceding cognitive decline that has been also suspected with CBF, is this concomitant with CBF in most people? This could maybe make CBF an easier and more straightforward biomarker since its effects mirror that of BBB? I suspect it generally does not, even in healthy aging. It would have been great to shed more light on this with your results and in your discussion.

Thank you for your comments. By 'distinct trajectory of kw changes with aging,' we refer to the ‘turning point’ in age at which kw starts declining. BBB kw remained relatively stable and began to decline in the early 60s, while CBF consistently decreased and ATT consistently increased with age, although the rates of change differed at 22 years and 36 years, respectively. Using linear regressions for voxel analysis, Figure 4 shows that age-dependent decreases in CBF and increases in ATT were observed in most of the brain. However, significant age-related decreases in kw were more localized to specific brain regions and were mostly accompanied by simultaneous decreases in CBF and increases in ATT. We highlighted this finding in the updated manuscript (L250): “In the brain regions showing significant age-related kw decreases (Fig. 4A), these decreases are mostly accompanied by CBF decreases (Fig. 4B) and ATT increases (Fig. 4C).”

Thank you for your suggestion regarding the relationship between kw and CBF. We further conducted linear regressions between regional kw and regional CBF or ATT, incorporating sex as a covariate, for participants aged 8-61 years and 62-92 years (when BBB kw starts declining). The results are summarized Supplemental Table S6.

This new supplemental tables shows many interesting results. BBB kw was significantly negatively associated with CBF in the putamen, amygdala, hippocampus, parahippocampal gyrus, and medial temporal lobe in participants younger than 62 years, when kw was relatively consistent across ages. However, no significant correlations were found in any brain regions in the 62-92 years group. In contrast to CBF, kw was significantly negatively associated with ATT in the GM, temporal lobe, and precuneus in participants aged 8-61 years, and these correlations became significant in additional ROIs, including WM, frontal lobe, ACC, caudate, putamen, amygdala, hippocampus, PHG, and MTL in participants aged 62-92 years.

We have added the following discussion to the updated manuscript (L307): 'We observed a distinct trajectory of kw changes with aging compared to CBF and ATT. To study the potential regional associations between kw and CBF and ATT, we conducted linear regressions between regional kw and regional CBF or ATT, incorporating sex as a covariate, for participants aged 8-61 years and 62-92 years (when BBB kw starts declining), respectively. The results are shown in Supplemental Table S6. BBB kw was significantly negatively associated with CBF in the putamen, amygdala, hippocampus, PHG, and MTL in participants aged 8-61 years (when kw was relatively consistent across ages), but no significant correlations were found in any brain regions in the 62-92 years group. In contrast to CBF, kw was significantly negatively associated with ATT in the GM, temporal lobe, and precuneus in participants aged 8-61 years, and these correlations became significant in additional brain regions, including WM, frontal lobe, ACC, caudate, putamen, amygdala, hippocampus, PHG, and MTL in participants aged 62-92 years. These results suggest that BBB function may be affected by different aspects of neurovascular function represented by CBF and ATT at different stages of aging."

Other notes:

R2.4 While reading the results section, two things that jump out at me when I saw the sex differences: (1) hematocrit and (2) menopausal status. I saw in the discussion that these were touched on. I may have missed this in the methods, was hematocrit collected and included in the parameters estimates?? Was the menopausal status including ERT (estrogen replacement therapies) recorded and factored in? If not these could be included as limitations that may confound the results, especially when the age groups were split to include a group comprising or potentially both pre-and post-menopausal females (36-61).

We do not have the information about hematocrit nor menopausal status and they were not included in data analysis. We agree this is a limitation of the current study and we discussed in the updated manuscript (L442): “Other factors such as hematocrit (23), menopausal status (24, 25), and vascular risk factors (26) should also be considered. These variables were not included in this study due to data unavailability or limited availability in some cohorts. We attempted to minimize the impact of these factors on our observations by including a relatively large and diverse sample. However, future studies examining the specific mechanism of each of these factors on BBB function in aging would be valuable.”

R2.5 The general vascular health of the cohort is not well described especially if some of the participants were from sickle cell study. While they are cognitively normal and free from major medical illnesses, or neurological disorders, did the sample also include individuals with considerable vascular risk factors and metabolic syndrome (known to affect CBF), especially in the older cohort??

We agree with the reviewer that vascular health can significantly impact perfusion and BBB function. Since the data presented in this study were collected from multiple cohorts, vascular risk factors were not available in all cohorts and thus were not included as covariates in the data analysis. To account for potential vascular variations across participants, we included CBF and ATT as covariates in our analysis on age related BBB kw changes. We have added discussion in the updated manuscript (L442, same as our response to the previous comment): “Other factors such as hematocrit (23), menopausal status (24, 25), and vascular risk factors (26) should also be considered. These variables were not included in this study due to data unavailability or limited availability in some cohorts. We attempted to minimize the impact of these factors on our observations by including a relatively large and diverse sample. However, future studies examining the specific mechanism of each of these factors on BBB function in aging would be valuable.”.

References:

(1) K. S. St Lawrence, D. Owen, D. J. Wang, A two-stage approach for measuring vascular water exchange and arterial transit time by diffusion-weighted perfusion MRI. *Magn Reson Med* 67, 1275-1284 (2012).

(2) X. Shao, C. Zhao, Q. Shou, K. S. St Lawrence, D. J. Wang, Quantification of blood–brain barrier water exchange and permeability with multidelay diffusion‐weighted pseudo‐continuous arterial spin labeling. *Magnetic Resonance in Medicine* (2023).

(3) P. Giannakopoulos, E. Kövari, F. R. Herrmann, P. R. Hof, C. Bouras, Interhemispheric distribution of Alzheimer disease and vascular pathology in brain aging. *Stroke* (2009).

(4) A. Mahroo, S. Konstandin, M. Günther, Blood–Brain Barrier Permeability to Water Measured Using Multiple Echo Time Arterial Spin Labeling MRI in the Aging Human Brain. *Journal of Magnetic Resonance Imaging* 59, 1269-1282 (2024).

(5) Y. Ohene *et al.*, Increased blood–brain barrier permeability to water in the aging brain detected using noninvasive multi‐TE ASL MRI. *Magnetic resonance in medicine* 85, 326-333 (2021).

(6) B. R. Dickie, H. Boutin, G. J. Parker, L. M. Parkes, Alzheimer's disease pathology is associated with earlier alterations to blood–brain barrier water permeability compared with healthy ageing in TgF344‐AD rats. *NMR in Biomedicine* 34, e4510 (2021).

(7) Y. Ying *et al.*, Heterogeneous blood‐brain barrier dysfunction in cerebral small vessel diseases. *Alzheimer's & Dementia* (2024).

(8) V. Zachariou *et al.*, Regional differences in the link between water exchange rate across the blood–brain barrier and cognitive performance in normal aging. *GeroScience*, 1-18 (2023).

(9) Y. Zhang *et al.*, Increased cerebral vascularization and decreased water exchange across the blood-brain barrier in aquaporin-4 knockout mice. *PLoS One* 14, e0218415 (2019).

(10) Y. Ohene *et al.*, Non-invasive MRI of brain clearance pathways using multiple echo time arterial spin labelling: an aquaporin-4 study. *NeuroImage* 188, 515-523 (2019).

(11) Y. V. Tiwari, J. Lu, Q. Shen, B. Cerqueira, T. Q. Duong, Magnetic resonance imaging of blood–brain barrier permeability in ischemic stroke using diffusion-weighted arterial spin labeling in rats. *Journal of Cerebral Blood Flow & Metabolism* 37, 2706-2715 (2017).

(12) Z. Wei *et al.*, Non-contrast assessment of blood-brain barrier permeability to water in mice: an arterial spin labeling study at cerebral veins. *NeuroImage*, 119870 (2023).

(13) Y. Jia *et al.*, Transmembrane water-efflux rate measured by magnetic resonance imaging as a biomarker of the expression of aquaporin-4 in gliomas. *Nature Biomedical Engineering* 7, 236-252 (2023).

(14) L. Nobis *et al.*, Hippocampal volume across age: Nomograms derived from over 19,700 people in UK Biobank. *NeuroImage: Clinical* 23, 101904 (2019).

(15) S. Rane *et al.*, Inverse correspondence between hippocampal perfusion and verbal memory performance in older adults. *Hippocampus* 23, 213-220 (2013).

(16) S. Heo *et al.*, Resting hippocampal blood flow, spatial memory and aging. *Brain research* 1315, 119-127 (2010).

(17) O. Gannon, L. Robison, A. Custozzo, K. Zuloaga, Sex differences in risk factors for vascular contributions to cognitive impairment & dementia. *Neurochemistry international* 127, 38-55 (2019).

(18) A. E. Leeuwis *et al.*, Cerebral blood flow and cognitive functioning in a community-based, multi-ethnic cohort: the SABRE study. *Frontiers in aging neuroscience* 10, 279 (2018).

(19) L. R. Clark *et al.*, Association of cardiovascular and Alzheimer’s disease risk factors with intracranial arterial blood flow in Whites and African Americans. *Journal of Alzheimer's Disease* 72, 919-929 (2019).

(20) D. R. Williams, S. A. Mohammed, Discrimination and racial disparities in health: evidence and needed research. *Journal of behavioral medicine* 32, 20-47 (2009).

(21) N. Scarmeas *et al.*, Association of life activities with cerebral blood flow in Alzheimer disease: implications for the cognitive reserve hypothesis. *Archives of neurology* 60, 359-365 (2003).

(22) N.-T. Chiu, B.-F. Lee, S. Hsiao, M.-C. Pai, Educational level influences regional cerebral blood flow in patients with Alzheimer’s disease. *Journal of Nuclear Medicine* 45, 1860-1863 (2004).

(23) R. C. Gur *et al.*, Gender differences in age effect on brain atrophy measured by magnetic resonance imaging. *Proceedings of the National Academy of Sciences* 88, 2845-2849 (1991).

(24) M. J. Cipolla, J. A. Godfrey, M. J. Wiegman, The effect of ovariectomy and estrogen on penetrating brain arterioles and blood-brain barrier permeability. *Microcirculation* 16, 685-693 (2009).

(25) A. C. Wilson *et al.*, Reproductive hormones regulate the selective permeability of the blood-brain barrier. *Biochim Biophys Acta* 1782, 401-407 (2008).

(26) M. S. Stringer *et al.*, Tracer kinetic assessment of blood–brain barrier leakage and blood volume in cerebral small vessel disease: Associations with disease burden and vascular risk factors. *NeuroImage: Clinical*
**32**, 102883 (2021).